# Synergistic Antimicrobial Activities of Thai Household Essential Oils in Chitosan Film

**DOI:** 10.3390/polym13091519

**Published:** 2021-05-09

**Authors:** Juthamas Tantala, Pornchai Rachtanapun, Chitsiri Rachtanapun

**Affiliations:** 1Department of Food Science and Technology, Faculty of Agro-Industry, Kasetsart University, Bangkok 10900, Thailand; tantala.j@gmail.com; 2Faculty of Agro-Industry, School of Agro-Industry, Chiang Mai University, Chiang Mai 50100, Thailand; pornchai.r@cmu.ac.th; 3The Cluster of Agro Bio-Circular-Green Industry (Agro BCG), Chiang Mai University, Chiang Mai 50100, Thailand; 4Center of Excellence in Materials Science and Technology, Chiang Mai University, Chiang Mai 50200, Thailand; 5Center for Advanced Studied Agriculture and Food, Kasetsart University, Bangkok 10900, Thailand

**Keywords:** active packaging, antimicrobial films, antimicrobial activity, biopolymers, chitosan film, essential oil, synergetic effect

## Abstract

Foodborne pathogens mostly contaminate ready-to-eat (RTE) meat products by post-process contamination and cause foodborne disease outbreaks. Preventing post-process contamination and controlling microbial growth during storage by packing the RTE meats with active antimicrobial film from chitosan combined with the synergism of Thai household essential oils was investigated. Here, we analyzed antimicrobial activity and mechanical properties of chitosan films incorporated with essential oil of fingerroot (EOF) and holy basil (EOH) based on their fractional inhibitory concentration and isobolograms. We showed that antimicrobial activities of chitosan film and chitosan films formulated with EOF:EOH displayed a dramatical reduction of *Listeria monocytogenes* Scott A concentration by 7 Log in 12 h. Chitosan film incorporated with EOF:EOH at ratio 0.04:0.04% *v*/*v*/*w* strongly retarded growth of total viable count of *L. monocytogenes* on vacuum-packed bologna slices during seven days of storage at 4 and 10 °C. Combined EOF and EOH added to chitosan films did not alter thickness, elongation (%) and colors (L*, a* and b*) of the chitosan film, but it increased water vapor transmission rate and decreased film tensile strength. Results suggested that chitosan film had strong antibacterial properties. Its effectiveness in inhibiting foodborne pathogenic bacteria in ready-to-eat meat products was enhanced by adding a combination of EOF:EOH.

## 1. Introduction

Active antimicrobial food packaging is an innovative concept that was introduced to respond to continuous changes in consumer demands and market trends. This technology illustrates its usage in food contact applications, which plays an important role in reducing the risk from pathogens and prolonging food shelf-life [1]. Regarding other biofunctional materials, chitosan has received significant interest to use as an antimicrobial film-forming agent due to its biodegradability, biocompatibility and antimicrobial activity. In our previous study, chitosan showed the strongest antilisterial activity, especially crab polymer chitosan [2]. Chitosan film is readily prepared by evaporation of its dilute acid solutions [3]. Nowadays, considerable interest has been created by the possibility of using chitosan film to retard or prevent the growth of foodborne and food spoilage microorganisms. This film serves as a carrier for a wide range of food additives including antimicrobial agents which act as a food preservative to prolong the shelf-life of products.

Antimicrobial efficacies of essential oils (EOs) were established long ago. EOs are hydrophobic liquids containing various volatile compounds extracted from several parts of plants [4]. Historically, EOs have been used in foods as flavorings, but they also have a broad spectrum against various microorganisms [5]. Several published studies have extolled the antimicrobial activity of EOs which occurs as a result of their ability to increase the permeability of bacterial cells, coagulate cytoplasm, reduce the intracellular ATP pool resulted to decrease ATP synthesis, damage cytoplasmic and membranes protein, alter the membrane of fatty acids and lower proton motive force across bacterial membrane [6]. Moreover, EOs were shown to disintegrate cell surfaces, detach the cytoplasmic membrane from the cell wall and change cytoplasmic inclusion by reducing vacuole size [7,8]. Numerous studies of the antimicrobial activity of various EOs and phytochemicals have been reviewed in great detail [9,10,11,12]. However, the requirement of a high concentration of EOs to enable sufficient inhibition of microorganisms in complex food systems is a difficult problem to tackle. This is due to their high volatility, strong odor and poor water solubility which affects food sensory quality. Thus, one of the possible ways to reduce the concentration of each EO required in the food matrix can be achieved through a combination of EOs with synergistic antimicrobial effects [13,14].

Previous studies examined synergistic antimicrobial effects; for example, carvacrol and linalool demonstrated synergistic effects against *L. monocytogenes* [15] and combined action of thymol/carvacrol, thymol/eugenol and carvacrol/eugenol revealed synergistic effects against *Escherichia coli* [16]. EOs from Thai household herbs such as holy basil, fingerroot, ginger, turmeric, lemongrass and black pepper have also been studied for their effects against pathogenic microorganisms and presented a broad spectrum against both Gram-positive and Gram-negative bacteria [17,18]. However, only a few reports have detailed the antimicrobial activity of their combinations. Moreover, incorporation of EOs by chitosan films as natural bactericides could enhance their antimicrobial effectiveness with potential development as environmentally friendly antimicrobial packaging. These films act as carriers to control the release of antimicrobials on the food surface which is the key factor in controlling microbial growth in products and thereby enhance product shelf-life.

*L. monocytogenes*, pathogenic *E. coli*, *Salmonella* spp. and *Staphylococcus aureus* are common foodborne pathogens associated with RTE meat and poultry products [19,20,21,22,23,24]. Among them, *L. monocytogenes* was the major concern on refrigerated RTE foods. It causes listeriosis which is widely regarded as one of the deadliest foodborne diseases and poses a serious threat to consumer health [25]. *L. monocytogenes* presents a high risk to processed RTE products because it can cross-contaminate from the processing environment to the food prior to packing and increase their numbers during storage in the refrigerator [26]. RTE meat products such as hot dogs, lunch meats, cold cuts and other deli meats (such as bologna) [27] are perishable and pose a higher risk for listeriosis infection in the event of post-process surface contamination such as a mechanical slicer [28,29,30]. Moreover, cold-eat RTE meat products that undergo no further control steps limited customers’ opportunities to eliminate the pathogen prior to food consumption. In the USA, an estimated 1600 people contract listeriosis annually, with about 260 deaths [25]. This problem has resulted from changes in consumer trends and legislation that prefer chemical preservatives free, safe, but minimally processed food [31,32,33]. For *S. aureus, E. coli* and *Salmonella* spp., these bacteria can cause food poisoning due to improper hygienic practices especially on the surface of equipment that is in contact with the food surface [26,34,35]. *Salmonella enterica subsp. enterica* serovar Typhimurium is often associated with many types of animals and has been associated with a wide range of foods including RTE meat products [36]. Moreover, *E. coli* and *Salmonella* isolated from RTE meat and poultry products were resistant to antibiotics, some isolates were multidrug resistant [19,35,37]. The prevalence of pathogenic bacteria in RTE meat can be substantially increased by post-cooking handling activities, exposure duration at points of sale and meat storage conditions [38].

In this study, we investigated the antimicrobial ability of EOs from Thai household herbs, including holy basil (EOH), fingerroot (EOF) and lemongrass (EOL) against foodborne pathogenic bacteria in a microbiological medium, both singly and in combination. Antimicrobial packaging films from crab polymer chitosan and chitosan films incorporated with selected EOs based on their fractional inhibitory concentration (FIC) index values were produced, and their antimicrobial activities were investigated on both laboratory broth and bologna as a food model. Physical and mechanical properties of the antimicrobial films were also studied.

## 2. Materials and Methods

### 2.1. Experimental Materials

Crab polymer chitosan (molecular weight of 700–1000 kDa and degree of deacetylation 97.38%) was purchased from Ta Ming Enterprises Co., Ltd. (Samut Sakhon, Thailand). Acetic acid and sodium chloride (NaCl) were obtained from Merck (Darmstadt, Germany). Sodium nitrite (NaNO_2_) was purchased from Lab Scan (Samut Sakhon, Thailand). For antimicrobial tests, trypticase soy agar (TSA), trypticase soy broth (TSB) and yeast extract (YE) were purchased from Merck (Darmstadt, Germany). Müeller Hinton broth (MHB), plate count agar (PCA) and peptone were bought from Difco (Maryland, USA). Modified *Listeria* selective agar base enrichment supplement (MOX) was sourced from Oxoid (Hampshire, England). Three types of Thai household essential oils including essential oil of fingerroot (EOF) (*Boesenbergia pandurata* (Roxb.) Schltr.), essential oil of holy basil (EOH) (*Ocimum sanctum* Linn.) and essential oil of lemongrass (EOL) (*Cymbopogon citratus* Stapf.) were purchased from Thai-China Flavours and Fragrances Industry Co., Ltd. (Pathum Thani, Thailand). The EOs were kept in aluminum bottles and stored at −20 °C until use [39].

### 2.2. Tested Bacteria

Foodborne pathogenic bacteria including *Escherichia coli* TISTR 780, *Salmonella enterica subsp. enterica* serovar Typhimurium ATCC 13,311 and *Staphylococcus aureus* TISTR 1466 were obtained from the Thailand Institute of Scientific and Technological Research, while *Listeria monocytogenes* Scott A was obtained from the Laboratory of Food Safety, Department of Food Science and Technology, Faculty of Agro-Industry, Kasetsart University, Thailand. *E. coli*, *S. enterica* and *S. aureus* were cultured in 10 mL of TSB, while *L. monocytogenes* was grown in 10 mL of TSB + 0.6% YE (TSBYE) and incubated at 37 °C for two successive 24 h and 18 h transfers before use.

### 2.3. Agar Dilution Assay

Minimum inhibitory concentrations (MICs) of EOs and their fractions were evaluated using the agar dilution method [40]. Antimicrobial activities of individual EOs were evaluated by stepwise-diluted concentrations. The checkerboard assay was also used to determine the combinations of tested EOs [41]. Test tubes each containing 13 mL of suitable melted agar were mixed with 2 mL of essential oil solution diluted in sterile water. The maximum concentration of tested essential oils in the dilution method was 1% *v*/*v*. After the addition of the EOs, the agar plates were poured and allowed to solidify. Tested microorganisms from Section 2.2 were diluted in 0.1% peptone water to ca. 7 Log CFU/mL. One microliter of bacterial culture was dropped on the agar using a micropipette (around 4 Log CFU/mL per spot) and allowed to dry. A control plate with only added sterile water was prepared and inoculated to ensure adequate growth of the tested microorganism. Then, the agar was incubated at 37 °C and bacterial growth was determined after 24 h. MIC is considered as the lowest concentration that completely inhibits growth [42]. All experiments were repeated in triplicate with each experiment duplicated.

### 2.4. Antimicrobial Interaction Assessment

MIC isobolograms: Antimicrobial interactions of mixed essential oil fractions were investigated by MIC values from the checkerboards assays which were plotted as MIC isobolograms as described by Davidson, Sofos and Branen [42]. Maximum concentrations of mixed essential oils were 1% *v*/*v*.

Fractional inhibitory concentration (FIC) index: The MICs of tested essential oils were transformed to FIC values as follows [43,44]:FIC_EOA_ = (MIC of EO_A_ in presence of EO_B_)/(MIC of EO_A_ alone)(1)
FIC_EOB_ = (MIC of EO_B_ in presence of EO_A_)/(MIC of EO_B_ alone)(2)

The FIC index was calculated from the FIC values of each essential oil:FIC_index_ = FIC_EOA_ + FIC_EOB_(3)

FIC_index_ results were interpreted as follows: FIC_index_ < 1, synergism; FIC_index_ = 1, additively; FIC_index_ > 1, antagonism [43].

### 2.5. Preparation of Chitosan Film and Chitosan Films Incorporated with Essential Oils of Fingerroot (EOF) and Holy Basil (EOH)

Chitosan forming solution (1% *w*/*v*) was prepared as described by Tantala J. et al. [45] and 20% sorbitol (*w*/*w* of chitosan) was added to chitosan solution as a plasticizer. After mixing, the chitosan solution was sterilized at 121 °C for 15 min. EOF and EOH were added to the chitosan solution at concentrations of 0.04:0.04 and 0.08:0.08% *v*/*v*/*w*. These concentrations were chosen based on FIC_index_ and 2-times FIC_index_ against *L. monocytogenes* from Section 2.4. The mixture was mixed by a homogenizer at 9500 rpm for 10 min under aseptic technique conditions. The plate pour method was used to cast all the film solution (21 g) on Petri dishes to form an antimicrobial film. Chitosan film without addition of EOs was used as control film. All films were dried under aseptic conditions at 30 °C for 24 h.

### 2.6. Evaluation of Viable Cell Count

The bactericidal activity of antimicrobial films was examined by using the microdilution as recommended by the National Committee of Clinical Laboratory Standards [46]. A flask contains 10 mL of *L. monocytogenes* ca. 7 Log CFU/mL and MHB + 0.6% yeast extract (MHBYE). Then, approximately 0.3 g of antimicrobial film was put in a flask. The suspension was incubated at 37 °C with shaking at 120 rpm. Samples of the suspension were taken at 0, 6, 12, 18 and 24 h, and the number of bacteria was examined using the spread plate technique. *L. monocytogenes* was grown on TSAYE. Then, the agar plates were incubated at 37 °C for 24 h and the number of colonies was counted. The control was the same condition on MHBYE without adding the film specimens. All experiments were repeated in triplicate with each experiment duplicated. Inhibition of bacterial growth was calculated as Log reduction of the cell number as follows:Log reduction = Log N_0_ − Log N (4)

Note: N_0_ = initial bacteria population

N = surviving bacteria population at each time point

### 2.7. Film Applications on Bologna

Bologna slices were bought from a local grocery store and the compositions (*w*/*w*) were 72% pork, 15% chicken, 9% water and 4% of seasoning. Uniform bologna slices (each slice was ca. 0.2 cm thick and 8.5 cm diameter, 15 g weight and a_w_ = 0.98) were stored at 4 °C until use but no longer than 1 day. *L. monocytogenes* Scott A from Section 2.2 was diluted in 0.85% NaCl to ca. 5 Log CFU/mL. Surface inoculation of the bologna slices was achieved by deposition of a sterile cotton swab spread on one side of each slice (final microbial concentration on bologna surface ca. 2 Log CFU/mL). Then, inoculated bologna slices were dried in a laminar flow for 30 min. Antimicrobial efficacies of films prepared from Section 2.5 were evaluated by placing either chitosan film or chitosan film + 0.04% EOs on the upper inoculated bologna surfaces. The bologna was placed in a sterile polyethylene plastic bag, then individually vacuum-sealed and finally stored at 4 or 10 °C for 1, 3, 5 and 7 days prior to enumerating bacteria using the spread plate technique. Inoculated bologna samples without application of the films served as controls. At the time of bacterial enumeration, the films were aseptically removed and bologna samples were mixed with 0.1% peptone for 2 min using stomacher. PCA was used for a total viable count (TVC), TSAYE overlaid with MOX was used for *L. monocytogenes*. All of the plates were incubated at 37 °C for 24 h before counting the colonies. Films recovered from the inoculated samples were flooded with 0.1% peptone water (pH = 6.3) with continuous stirring for 5 min at room temperature. Bacteria in the wash peptone water were performed by spread plate using PCA for TVC and TSAYE for *L. monocytogenes*. The plates were incubated at 37 °C for 24 h before counting the colonies. This experiment was done in duplicate.

### 2.8. Film Thickness

Film thickness was measured with average measurement from five random locations on the films by using a microcaliper (Mitutoyo, Japan). The mean thickness values were used to calculate the mechanical properties of films including tensile strength, percentage elongation and water vapor transmission rate.

### 2.9. Tensile Strength (TS) and Percentage Elongation (%E)

Tensile strength (TS) and percentage elongation (%E) were studied according to method ASTM-D882 [47], with some adaptation [48] using a texture analyzer (TAXT plus; Microsystem, England). Before testing, film strips (25 mm × 150 mm) were equilibrated at 65% relative humidity and 25 °C for 48 h. Initial grip separation and cross-head speed were set at 100 mm and 20 mm/min, respectively. Testing of each film’s specimen and calculation of TS and %E was conducted as describe by Rachtanapun and Tongdeesoontorn [48]. The TS value was calculated by dividing the maximum load with the initial cross-sectional area of the specimen. The %E value was calculated by dividing the elongation at the moment of rupture of the sample by the initial gauge length of a specimen (150 mm) of the sample and multiplied by 100.

### 2.10. Water Vapor Transmission Rate (WVTR) of Films

Water vapor transmission rate (WVTR) of the film sample was measured using the method ASTM E96 [49] at 65% relative humidity (using a saturated solution of NaNO_2_). The calculations of WVTR values were performed as described by Rachtanapun [50]. WVTR (g·m^−^^2^·d^−^^1^) was calculated from the slope of weight gain versus time per area of film sample.

### 2.11. Color Measurement

All the antimicrobial film samples were color measured using Mini Scan XE. Films were placed in glass Petri dishes and results were reported as L*, a* and b* values based on International Commission on Illumination (CIE) Lab values All samples were examined in triplicate.

### 2.12. Statistical Analysis

The experimental study design for antimicrobial chitosan film properties and shelf-life of bologna samples were conducted by using complete randomized design (CRD). Analysis of variance (ANOVA) was performed on all results using SPSS version 12.0 (SPSS, Inc.; IBM Corp.; Chicago, IL, USA). at a confidence interval of 95% to determine significant differences between group samples.

## 3. Results and Discussion

### 3.1. Minimum Inhibitory Concentration (MIC) of Essential Oils

Three EOs showed significant antibacterial activities depending on the type of tested bacteria as presented in Table 1. Essential oil of fingerroot (EOF) show the strongest antibacterial against *S. aureus* (MIC = 0.10% *v*/*v*), while essential oil of holy basil (EOH) and essential oil of lemongrass (EOL) showed the strongest against *L. monocytogenes* and *S. aureus* (MIC = 0.10% *v*/*v*). Gram-positive bacteria (*L. monocytogenes* and *S. aureus)* were more susceptible to EOs than Gram-negative bacteria *(E. coli* and *S. enterica*), in agreement with our previous studies [5,6,13,17]. A reduction in Gram-negative bacteria sensitivity to EOs is likely caused by the protection from the hydrophilic lipopolysaccharide in their outer membrane, which then prevents EOs entering and damaging the bacterial cell [51]. A small reduction in the MIC is likely caused by permeation of EOs through large hydrophilic pores protein in the outer membranes.

### 3.2. A Screen of Antimicrobial Combination Effect

We screened series of antimicrobial synergistic effects using MIC isobolograms studies. Using this method, we calculated an MIC for two antimicrobial agents at three different concentrations. If a combination of two antimicrobial agents yields higher inhibition than a single concentration alone, then two antimicrobial agents are acting synergistically (Figure 1). Synergistic effects were observed between EOF and EOH against all tested bacteria, EOH and EOL against *E. coli* and *S. enterica*, and EOF and EOL against *S. enterica*, while EOF and EOL recorded isobolograms showing antagonism against both Gram-positive bacteria and *E. coli*.

To simplify our calculations, we calculated the FIC_index_ using mathematical terms as described in materials and methods. We show that the combination of EOF and EOH interacted synergistically against all tested bacteria with FIC_index_ ranging 0.45–0.80 (Table 2). Synergistic interactions were also found in combinations between EOH and EOL against *E. coli* and *S. enterica* with FIC_index_ at 0.53 and 0.60, respectively and a combination between EOF and EOL against *S. enterica* with FIC_index_ at 0.70. However, an antagonistic effect was found in EOF and EOL against both Gram-positive bacteria and EOF and EOL against *L. monocytogenes*, *S. aureus* and *E. coli* with FIC_index_ > 1. The generated FIC_index_ was confirmed in substantial agreement with the antimicrobial combination interactions from the MIC isobolograms, which showed interaction of EOF, EOH and EOL against foodborne pathogenic bacteria.

The synergistic effect of two agents can occur in several types of situations and are closely related to the antimicrobial efficacy of the dominant compounds which comprise each EO [5]. Camphor (30.34%) and 1,8-cineole (20.25%) were found to be the main components in EOF [7]. EOs containing high amounts of 1,8-cineole showed stronger antilisterial activity than other essential oils [52]. In EOH, the strongest antimicrobial activity was recognized to the components of eugenol (41.5%) and methyl-eugenol (11.8%). This is likely due to the potent antimicrobial hydroxyl group (–OH) on phenols [53]. The main component of EOL was determined as citral (44.6%) which showed antibacterial activity against *L. monocytogenes* [52] and *E. coli* [54]. Combinations of EOF:EOH, EOH:EOL and EOF:EOL against all tested bacteria were attributed to the mode of action of their main components. Eugenol permeabilizes the cell membrane and interacts with proteins to increase potassium and ATP cellular efflux [55,56] while 1,8-cineole acted on the plasma membranes [57] and the mode of action of citral was membrane disruption and damage [54]. Therefore, the interaction of synergistic or antagonistic effects may be due to their target action on bacterial cells and the bacteria cell type and their outer membrane composition.

### 3.3. Antimicrobial Activity of Chitosan Films Combined with EOF and EOH in a Culture Broth Medium

Crab chitosan, as a natural biodegradable high molecular weight polymer was selected to cast the antimicrobial films based on its vigorous antilisterial activity among chitosan groups from our previous study [2]. Effectiveness of the chitosan films combined with EOF and EOH at different concentrations was assessed in a suitable broth medium against *L. monocytogenes* at the initial ca. 7 Log CFU/mL. Reduction of cell population from the antimicrobial film was conducted using a viability test and inhibition of bacterial growth was reported as Log reduction of the cell number. Chitosan films inactivated *L. monocytogenes* within 12 h by reducing initial cells at 7.0 Log CFU/mL to an undetectable level (Figure 2). Addition of EOF and EOH to the chitosan films showed a stronger antilisterial effect. The chitosan film + 0.08% EOs achieved 7 Log reduction at 6 h, reflecting the enhancement of antimicrobial film efficacies. This technology supports previous application as shown by Kanatt et al. [58] who reported that the addition of mint extract into chitosan film completely inhibited the number of tested bacteria including *E. coli*, *Bacillus cereus* and *S. aureus* within 4 h.

### 3.4. Application of Chitosan Film Combined with Essential Oils from EOF and EOH on Bologna

Based on the results of synergistic effects obtained from both MIC isobolograms and the FIC_index_, *L. monocytogenes* was chosen as a target bacterium for RTE deli meat to assess the antibacterial activity of active chitosan film. The film was added with a combination of EOF and EOH (EOF:EOH) at concentration ratios of 0.04:0.04 (chitosan film + 0.04% EOs) and 0.08:0.08 (chitosan film + 0.08% EOs)%*v*/*v*/*w*, respectively. Their antimicrobial efficacies on bologna against *L. monocytogenes* were examined using total viable counts (TVC). The microbiological criteria of bologna samples followed the Thai Industrial Standards Institute [59] and stipulated that TVC shall not be higher than 6 Log CFU/g.

Survival numbers of bacteria on bologna stored at temperatures 4 and 10 °C are shown in Figure 3. At 4 °C, the number of *L. monocytogenes* and TVC for all treatments showed negligible change because low temperatures retarded the growth of bacterial cells. By contrast, for storage at 10 °C, both *L. monocytogenes* and TVC increased rapidly in the bologna slices without antimicrobial films (control) and number of TVC was higher than the acceptable level at day 5. It can be assumed that the shelf-life of the control sample stored at 10 °C was less than 5 days. Chitosan film showed an antilisterial effect on bologna slices and retarded the growth of *L. monocytogenes*. However, the film did not show much inhibition effect on TVC. Bacteria present as TVC on the refrigerated bologna were possibly other spoilage bacteria such as *Brochothrix thermosphacta*, *Pseudomonas*, *Moraxella* and lactic acid bacteria [60]. Some of these spoilage bacteria, are Gram-negative bacteria which are more resistant to chitosan and EOs than Gram-positive bacteria.

Shelf-life of bologna slices covered with chitosan film was extended to 6 days. Interestingly, chitosan films + 0.04% EOs retarded growth of *L. monocytogenes* and TVC throughout the storage due to the bacteriostatic of this film. Thus, a combination of EOF:EOH in concentration ratio at 0.04:0.04% *v*/*v*/*w* in chitosan film was more effective at inhibiting the growth of TVC than plain chitosan films. EOF and EOH slightly improved the antimicrobial effect of chitosan film. When chitosan is solid, its antimicrobial activity is only located on the surface of the film [61]. Thus, the growth of microorganisms on bologna surface is only inhibited when the bacteria grow to make contact with the chitosan itself [33]. However, when antimicrobial agents such as EOs were added to the films, this allows EOs to diffuse from the chitosan surface. This almost allows the film to inhibit bacterial in two ways, both at the surface and at the distance away from the film [33]. This agrees with Zivanovic et al. [62], who reported that chitosan film incorporated with oregano essential oil improved the bactericidal effect of chitosan.

Chitosan films with and without EOs were analyzed for surviving bacteria consist of TVC and *L. monocytogenes* using the rinse test followed by the spread plate technique. Results showed that no viable cell counts of tested bacteria were detected (detection limit = 10 CFU/g) for all film samples during storage time (data not shown).

### 3.5. Mechanical Properties

The mechanical properties of antimicrobial films were assessed by measuring their tensile strength (TS) and percentage elongation (%E). Results showed that adding EOF and EOH to chitosan films resulted in a change in the properties and reduction of TS (Table 3). The TS of chitosan films decreased from 14.78 to 7.88 MPa when added EOF:EOH at concentrations of 0.04:0.04% *v*/*v*/*w*. Similar to previous reports presented that TS of the chitosan films decreased when added garlic oil, potassium sorbate, nisin or phenolic acids to chitosan films [39,63]. Lowering of TS value occurred due to a decrease in intermolecular crosslinking of chitosan matrix when incorporated with additives other [63]. Moreover, the addition of EOF:EOH at the concentration of 0.04:0.04 % *v*/*v*/*w* did not induce any significantly affect (*p* ≤ 0.05) in %E values in chitosan films. The change in mechanical properties of the films with the incorporation of EOF and EOH may occur due to the interaction between the chitosan matrix and phenolic compounds from EOs [63,64]. In phenolic acids-chitosan composited films, phenolic acids in the chitosan film interrupt the formation of ordered crystal structure, weakens intermolecular hydrogen bonds and obstructs the polymer–polymer chain interaction, which decreased the mechanical properties of chitosan film. Furthermore, the structure of phenolic acids also caused the difference in mechanical properties of chitosan film. Among gallic acid, ferulic acid, vanillic acid and salicylic acid, gallic-chitosan composite films possessed the best mechanical properties due to three phenolic hydroxyl groups in its molecule [63].

### 3.6. Water Vapor Transmission Rate (WVTR)

Water vapor transmission rate (WVTR) is a measure of how easily moisture penetrates and passes through a material [39]. The addition of EOF and EOH significantly increased (*p* ≤ 0.05) WVTR of chitosan film (Table 3). In general, the addition of antimicrobial agents increased WVTR value because they contributed to extending intermolecular interaction and loosened the compactness of the structure. This phenomenon induced segmental motions and free space, causing a more open matrix and enhanced moisture passing through the edible films, thereby increasing WVTR values [39].

### 3.7. Color Measurement

All films in this experiment showed very similar color characteristics. The L* and a* values of all films were not significantly different (*p* ≤ 0.05). Pure chitosan film was used as a color reference for chitosan film incorporated with EOF:EOH at concentration 0.04:0.04% *v*/*v*/*w*. The appearance of chitosan film without EOs was slightly yellow and transparent. Film transparencies reduced as amounts of EOF and EOH increased. The addition of EOF and EOH gave an intense yellow color with non-transparent characteristics as shown in Figure 4. Therefore, the incorporation of EOF and EOH into these films led to moderate changes in their appearances.

## 4. Conclusions

The synergistic effects of EOF and EOH successfully inhibited tested bacteria in vitro. Incorporation of essential oils from EOF and EOH based on FIC_index_ in chitosan successfully developed active antimicrobial films. Combining EOF and EOH in chitosan film increased WVTR and %E but decreased TS and film transparencies. Enhancing the antimicrobial activity of chitosan films by adding a combination of EOF and EOH reduced the number of *L. monocytogenes* from 7 Log CFU/mL to an undetectable level within 12 h. Application of chitosan film incorporated with EOF and EOH at 0.04:0.04%*v*/*v*/*w* showed stronger inhibitory effect against TVC and *L. monocytogenes* than chitosan film during storage times. Therefore, chitosan films incorporated with EOF and EOH at 0.04:0.04%*v*/*v*/*w* had the potential to inhibit foodborne pathogens in RTE meat products. It could be applied as an edible antimicrobial film covered on the cold cuts or deli meats after slicing which can provide as a carrier distributing antimicrobials onto the food surface to control growth of post-process contaminated foodborne pathogens and extending the shelf-life of products. Further work is currently on-going to evaluate other important issues, such as sensory evaluation and water resistance of films to ensure their ability to be applied in RTE meat processors.

## Figures and Tables

**Figure 1 polymers-13-01519-f001:**
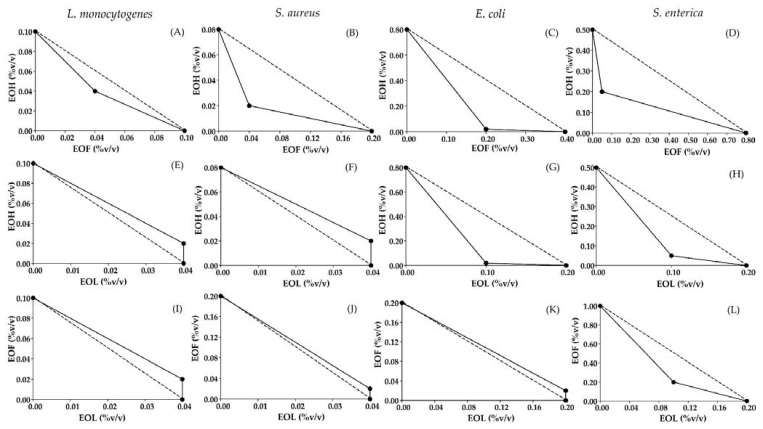
(**A**–**L**) Minimum inhibitory concentration (MIC) isobolograms of EOF, EOH and EOL combinations against foodborne microorganisms. Straight lines (dotted line) represent the reference additive lines and points correspond to mean MIC data from checkerboard assay. On the MIC isobolograms, additive interactions are represented as a straight line, while the variation of linearity to the right or left of the additive line is interpreted as antagonism or synergism, respectively [42].

**Figure 2 polymers-13-01519-f002:**
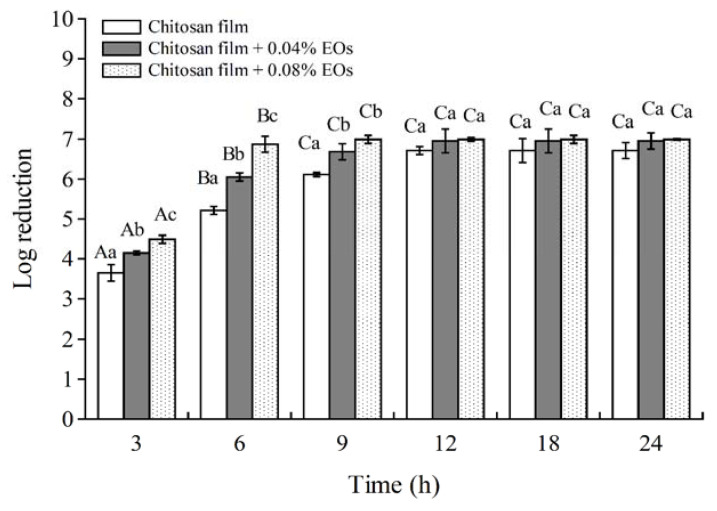
Antilisterial activity of chitosan film incorporated with EOF:EOH at 0.04:0.04 (Chitosan film + 0.04% EOs) and chitosan film incorporated with EOF:EOH at 0.08:0.08 (Chitosan film + 0.08% EOs). Error bars represent ± standard deviation from three replicates. Bars of the same treatment, followed by different capital letters indicate significant differences (*p* ≤ 0.05). Bars of the same time followed by different small letters indicate significant differences (*p* ≤ 0.05).

**Figure 3 polymers-13-01519-f003:**
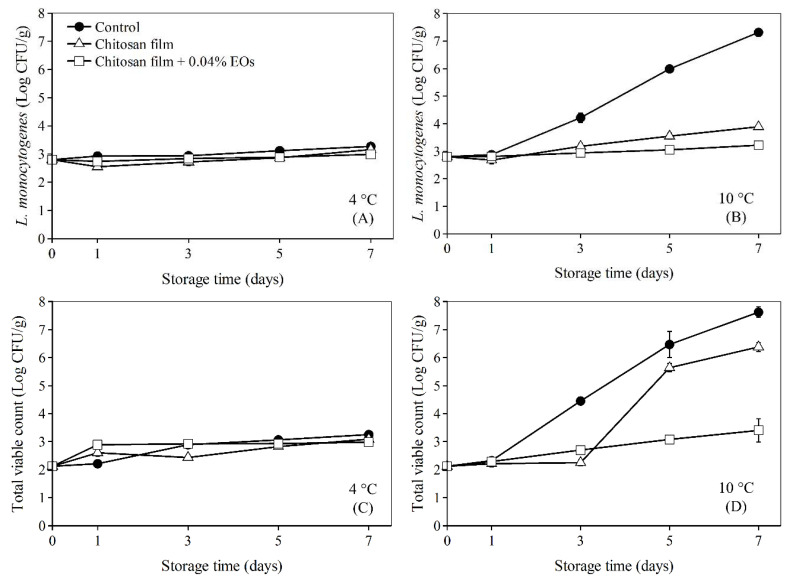
Number of *L. monocytogenes* at 4 °C (**A**), 10 °C (**B**) and total viable counts at 4 °C (**C**), 10 °C (**D**) on bologna stored for seven days. Error bar represents standard deviation over three repeats.

**Figure 4 polymers-13-01519-f004:**
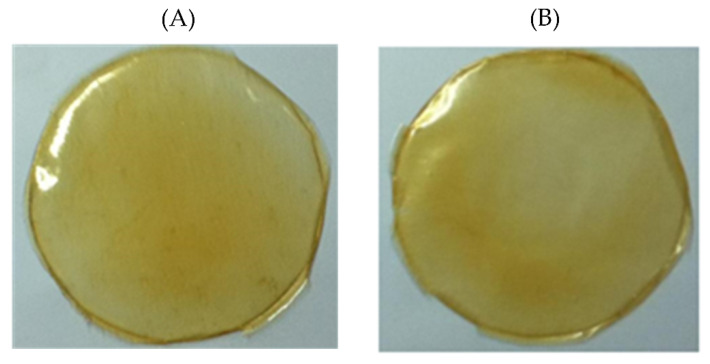
Antimicrobial film from chitosan (**A**) and chitosan film + 0.04% EOs (**B**).

**Table 1 polymers-13-01519-t001:** Minimum inhibitory concentrations (MICs) of various essential oils against *L. monocytogenes*, *S. aureus*, *E. coli* and *S. enterica* determined by agar dilution method.

Bacterium	MIC (%) of Essential Oils		
	Fingerroot	Holy Basil	Lemongrass
*L. monocytogenes*	0.30	0.10	0.10
*S. aureus*	0.10	0.10	0.10
*E. coli*	0.40	0.50	0.20
*S. enterica*	0.40	0.50	0.50

**Table 2 polymers-13-01519-t002:** Fractional inhibitory concentration index (FIC_index_) of EOF, EOH and EOL against *L. monocytogenes*, *S. aureus*, *E. coli* and *S. enterica* ^a^.

Microorganism	Ratio of Essential Oil Combinations		FIC			FIC_index_
			EOF	EOH	EOL	
*L. monocytogenes*	EOF:EOH	0.50:0.50	0.04	0.04	-	0.80(S) ^b^
	EOH:EOL	0.33:0.67	-	0.20	0.10	1.20 (AN)
	EOF:EOL	0.33:0.67	0.20	-	0.10	1.20 (AN)
*S. aureus*	EOF:EOH	0.33:0.67	0.20	0.10	-	0.45(S)
	EOH:EOL	0.33:0.67	-	0.25	0.10	1.25 (AN)
	EOF:EOL	0.33:0.67	0.10	-	0.10	1.10 (AN)
*E. coli*	EOF:EOH	0.09:0.91	0.50	0.30	-	0.53 (S)
	EOH:EOL	0.17:0.83	-	0.03	0.50	0.53 (S)
	EOF:EOL	0.09:0.91	0.10	-	1.10	1.10 (AN)
*S. enterica*	EOF:EOH	0.80:0.20	0.06	0.40	-	0.46 (S)
	EOH:EOL	0.33:0.67	-	0.10	0.50	0.60 (S)
	EOF:EOL	0.67:0.33	0.20	-	0.50	0.70 (S)

^a^ FIC value was calculated by dividing MIC of an antimicrobial when used in combination with MIC of the antimicrobial when used alone. FIC was the sum of FIC values of the two antimicrobials used in combination. ^b^ The antimicrobial effect of combinations was synergistic (S, FIC < 1), additive (A, FIC = 1) or antagonistic (AN, FIC > 1).

**Table 3 polymers-13-01519-t003:** Mechanical and physical properties of chitosan film and chitosan film incorporated with EOF:EOH at 0.04:0.04 (Chitosan film + 0.04% EOs). Values are mean ± standard deviation. Different letters in the same column indicate significant differences between the means using *t*-test (*p* ≤ 0.05).

TYPE OF FILM	Mechanical Properties				Color		
	Thickness (mm)	Tensile Strength (MPa)	Elongation at break (%)	WVTR (g·m^−^^2^·d^−^^1^)	L*	a*	b*
Chitosan	0.10 ± 0.01 ^a^	14.78 ± 2.71 ^b^	13.49 ± 6.59 ^a^	16.22 ± 0.62 ^a^	25.24 ± 3.52 ^a^	−1.81 ± 0.21 ^a^	2.91 ± 0.75 ^a^
Chitosan film + 0.04% EOs	0.11 ± 0.01 ^a^	7.88 ± 1.54 ^a^	5.61 ± 0.55 ^a^	25.88 ± 5.19 ^b^	26.51 ± 6.38 ^a^	−1.90 ± 0.56 ^a^	4.24 ± 2.01 ^a^

## Data Availability

The data presented in this study are available on request from the corresponding author.

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
