# Peer review of "Synergistic Antimicrobial Activities of Thai Household Essential Oils in Chitosan Film"

_polymers, 2021, doi:10.3390/polym13091519_

Round 1

Reviewer 1 Report

The manuscript entitled “Synergistic antimicrobial activities of Thai household essential oils in chitosan film” have demonstrated the antimicrobial activity of essential oils (EOs) from Thai herbs like holy basil, fingerroot and lemongrass against foodborne pathogens (E.coli, Salmonella, S. aureus and L. monocytogenes). Moreover antibacterial interactions of mixed Eos fractions were investigated. Authors have investigated antimicrobial properties of packaging films from crab polymer chitosan and chitosan films with selected EOs and its physical and mechanical properties.

The study was written carefully and well in terms of language. The proposed statistical methods were appropriately selected according to the experiment and allow for the proper analysis of the research results. Authors should correct manuscript according to the suggestion.

Minor issues:

Materials and methods

Line 123 – 124: Authors have dissolved EOs with agar medium without any compounds that reduce surface tension e.g. Tween?

References:

Authors should checked and corrected (journal name) Reference no 39 and 44 according to journal guidelines

Reviewer 2 Report

The manuscript from Tantala et al. describes the development of chitosan films incorporated with antimicrobial essential oils. The authors first evaluated the antimicrobial action of these oils, individually and in combination, agains foodborne pathogens. After the formulation of the films, their mechanic and physical characteristics were determined. 

Overall, the paper is well written and organized and deserve be accepted to publication in this journal, after minor revisions. 

The suggestion can be found in the attached pdf file. 

Reviewer 3 Report

Generally, the manuscript is well written and studies also were well planned. Only a few points should be clarified by the Authors.

Remarks:

  1. "Melted agar were mixed with 2 mL of essential oil solution dissolved in sterile water as diluents." Did essential oils really dissolve in water, which is a polar solvent? Please explain. (Lines 123-124, page 3)
  2. Please specify what do exactly mean the expressions - "MIC of A or MIC of B? (Lines 142-143, page 3)
  3. Tensile strength and water vapor transmission rate of film samples were calculated according to the specific formulas. Please provide these formulas. (page 5)
  4. Could the Authors briefly describe the MIC results obtained for the tested etheric oils? Which etheric oil was the most effective in relation to the tested bacterial strain? (Lines 228-235, page 5)
  5. It is a pity that the Authors describing the synergistic and antagonistic effects of the etheric oils used, did not include their chemical composition. Maybe Authors have such data and they could include it in the manuscript.
  6. Which polyphenolic compounds from the etheric oils may have interacted with chitosan matrix? Please list them and briefly discuss. (Lines 386-388, page 10)
  7. Did the changes in the appearance of the foil after adding etheric oils made it more visually appealing? Could the Authors present some pictures? (Lines 406-409, page 11)
  8. In References part, Latin names of plants and names of bacterial strains should be written in italics. Besides, the names of journals should be written in capital letters.
